# Development and Application of Earth Observation Based Machine Learning Methods for Characterizing Forest and Land Cover Change in Dilijan National Park of Armenia between 1991 and 2019

Nathalie Morin [1], Antoine Masse [1], Christophe Sannier [1,*], Martin Siklar [2], Norman Kiesslich [2], Hovik Sayadyan [3], Loïc Faucqueur [1] and Michaela Seewald [2]

[1]   Collecte Localisation Satellites, 61 rue de la Cimaise, Green Park–Bâtiment C, 59650 Villeneuve d'Ascq, France; nmorin@groupcls.com (N.M.); amasse@groupcls.com (A.M.); lfaucqueur@groupcls.com (L.F.)
[2]   GeoVille Information Systems and Data Processing GmbH, Sparkassenplatz 2, 6020 Innsbruck, Austria; siklar@geoville.com (M.S.); kiesslich@geoville.com (N.K.); seewald@geoville.com (M.S.)
[3]   United Nations Development Programme, 14 Petros Adamyan Street, Yerevan 0010, Armenia; hovik.sayadyan@undp.org
*   Correspondence: csannier@groupcls.com

**Abstract:** Dilijan National Park is one of the most important national parks of Armenia, established in 2002 to protect its rich biodiversity of flora and fauna and to prevent illegal logging. The aim of this study is to provide first, a mapping of forest degradation and deforestation, and second, of land cover/land use changes every 5 years over a 28-year monitoring cycle from 1991 to 2019, using Sentinel-2 and Landsat time series and Machine Learning methods. Very High Spatial Resolution imagery was used for calibration and validation purposes of forest density modelling and related changes. Correlation coefficient $R^2$ between forest density map and reference values ranges from 0.70 for the earliest epoch to 0.90 for the latest one. Land cover/land use classification yield good results with most classes showing high users' and producers' accuracies above 80%. Although forest degradation and deforestation which initiated about 30 years ago was restrained thanks to protection measures, anthropogenic pressure remains a threat with the increase in settlements, tourism, or agriculture. This case study can be used as a decision-support tool for the Armenian Government for sustainable forest management and policies and serve as a model for a future nationwide forest monitoring system.

**Keywords:** forest density; forest degradation; deforestation; land cover change; Sentinel-2; landsat; very high spatial resolution imagery; machine learning

## 1. Introduction

### 1.1. Context

The Republic of Armenia (RA) is located at the junction of the biogeographic zones of the Lesser Caucasus and the Iranian and Mediterranean zones and exhibits both a great range of altitudinal variation from 375 m to the 4090 m peak of Mount Aragats, as well as a diversity of climatic zones. This combination has resulted in a variety of landscapes and ecological communities with a distinct flora and fauna, including many regionally endemic, relict, and rare species. Therefore, in the country's small territory of 2,974,300 ha, there are about 3800 species of high vascular plants, 428 species of soil and water algae, 399 species of mosses, 4207 species of fungi, 464 species of lichens, 549 species of vertebrates and about 17,200 species of invertebrates [1].

The first Forest Code of Independent RA that was accepted in 1994 was rather conservative and logically connected to the former Armenian Soviet Socialist Republic (SSR)

Forest Code established in 1978. According to that law, any production loggings in Armenian forests were forbidden and only certain so-called sanitary cuttings such as the removal of dead and infected trees were allowed. In 2005, the new Forest Code was approved where protection and special categories for production forest (along with state border, green zones, etc.) were introduced. Despite the clear indications of limited regular logging potential from Armenian forests, illegal and unregulated loggings with different intensities, which started in 1992 due to transportation and energetic blockade of the country, continue to this day [2].

A network of specially protected areas (SPAs) was first established in Armenia in 1958 to protect ecosystems and habitats as well as rare, endemic, and threatened species. The first specially protected nature areas of Armenia were Dilijan, Khosrov Forest and Shikahogh Reserves. The same year, six reservations were also established. All of them were of forest protection significance. Later, the network of SPAs was extended: in 1981, Erebuni Reserve was established in the vicinity of Yerevan to protect wild-growing cereals; in 1987, Sev Lich ("Black Lake" in Armenian) Reserve was established to protect the natural complex of the relict volcanic lake and in 1978, Sevan National Park was established based on the extremely important task to conserve fresh-water resources of Lake Sevan for the whole Transcaucasus, as well as on the national significance of fish resources [3].

In 1999, there were five State Reserves, 22 State Reservations and one National park registered with 311,000 ha or 10% of the surface of the RA. Around 60% of Armenian species are represented within the protected area network, however there is a bias towards forest habitats, and a need to expend the system to include better representation of other ecosystems [4].

Currently, SPAs are implemented under four different national designations: (1) state reserves (Khosrov, Shikahogh and Erebuni); (2) national parks (Sevan, Dilijan, Lake Arpi and Arevik); (3) state preserves (a total of 27 state preserves); and (4) national monuments (a total of 232 natural monuments). The total area of nationally designated protected areas covers 12.90% of the terrestrial territory and inland waters of RA. However, some efforts are still needed to reach the Aichi biodiversity target 11, which aims to have 17% of national territory protected [1].

Several decisions have been elaborated and approved by the Government of the RA for the improvement of legislation related to specially protected nature areas and the protection of flora and fauna objects. In particular: (a) The RA Government adopted decree No. 781-N "on Establishing the procedure of utilization of items of flora for their protection and reproduction in natural conditions" in 2014, which defines measures to protect the newly detected species registered in the Red Data Book of Armenia, including the delineation of the protection zones and the limitation of some of the economic activities there; (b) "The management plan and priority measures of "Dilijan" national park for 2017–2026" were approved according to the Government Resolution N 190-N in 2017 [5].

Dilijan National Park (NP) SNCO was established on 21/02/2002 by N 165 decree of the RA Government, on the base of "Dilijan" state reserve, set up by N P-341 decree of ArmSSR Council of Ministers. It is managed by the "Dilijan" NP administration, which is subordinate to the RA Ministry of Environment [6] (p. 89).

Although many studies on the monitoring of land cover/land use and forest degradation have been conducted at a global scale [7,8], regional or country level [9–11], there are no studies available over Armenia for the mapping of specially protected areas, using Earth Observation 5EO) data and remote sensing techniques.

### 1.2. Aim and Objectives

The objective of the present paper is to assess and report on the state of Armenia's forests with a focus on the Dilijan National Park (See Figure 1). This park is one of the four National Parks in Armenia, located in the Northern part of Armenia on the slopes of the Pambak, Miapor and Aregani mountain chains and on the basin of Aghstev and Getik rivers. Although the area became a specially protected area in 1958 in order to protect the

park's flora and fauna, the National Park was only established in the year 2002. Today, the Dilijan National Park is known for its forest landscapes, rich biodiversity, medicinal mineral water springs, natural and cultural monuments, and extensive network of hiking trails. We proved the suitability of validated EO products for an independent forest monitoring and support land use planning for sustainable forest management. Two main tasks have been performed: (1) providing detailed land use and land cover classification maps for the epochs 1991–2019 as a basic planning support tool for characterizing forest ecosystems and main evolution trends and drivers of deforestation in Dilijan National Park, Armenia, and (2) delivering an in-depth examination of the forest classes of the previous service, subdividing forested areas into different forest density and type classes. The service shall focus on assessing and locating areas where deforestation or forest degradation is taking place.

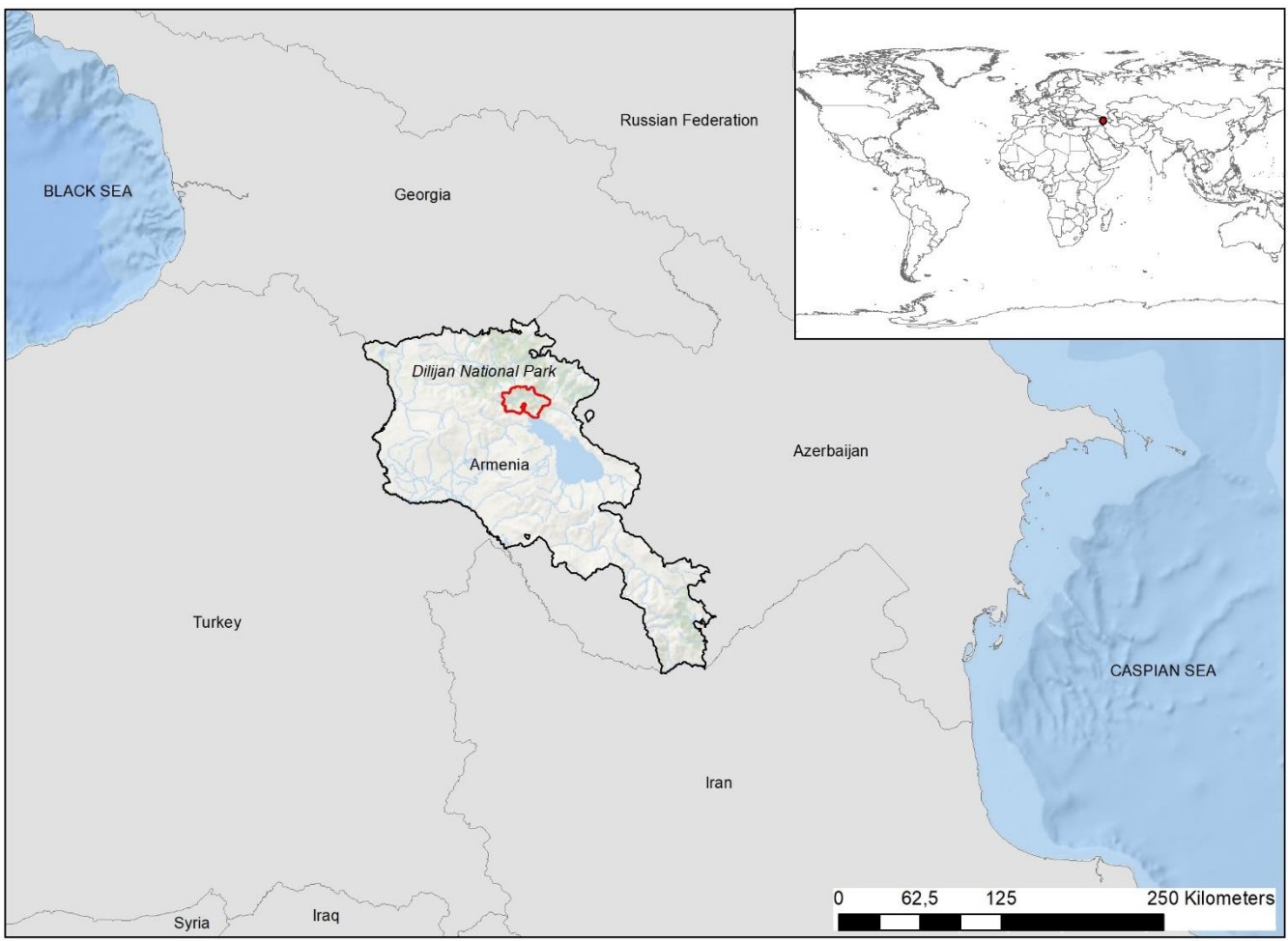

**Figure 1.** Study area: Dilijan National Park, Armenia ©OpenStreetMap (2021), ESRI World Oceans.

## 2. Data

### 2.1. Study Area

Armenian forests are among the most threatened ecosystems, with degradation accelerating, largely attributable to deforestation and overexploitation. This results in high rates of erosion, increasing soil salinity, lowered soil fertility, and loss of biodiversity. Thus, "expansion of forests is one of the main goals for Armenia, not only for the forests' protective role, but also to develop forest-related businesses and to ensure fuelwood supply to the population", said Ekrem Yazici, Deputy Chief of the Joint UNECE/FAO Forestry and Timber Section. However, due to a dense population living within the protected forest areas, developed infrastructures, uncontrolled tourism, illegal logging, poaching and non-sustainable

use of natural resources, environmental degradation threatens this unique biodiversity and the rich natural-historical and cultural landscapes. To become more resilient to external and internal shocks, there is a necessity for the integration of new approaches and policy instruments to rehabilitate degraded forests and increase forest cover significantly, while formulating the country's development agenda. In this context, UNDP Armenia is focusing its efforts to better understand the past forest ecosystems transformations, the land use and land cover changes, and in general, all the socio-environmental processes in the past and today that affect the sustainable management of forest resources.

### 2.2. In Situ Data

Armenian counterparts have provided three sets of data sources named: (a) vector data on basic cartographic layers–Dilijan NP border, hydrography, roads and railways, settlements, forest ecosystems status data on compartment level (species, density, site class, age group), red listed species (fauna and flora); (b) raster data–AlosPalsar 12 m resolution digital terrain model from 2007 and high resolution satellite images (Pléiades) for 2015, 2018–2019 years and (c) cadastral maps for the 2013–2015 period of seven communities in Dilijan NP. All the vector data, Alos Palsar raster data and cadastral maps of communities were provided by the Ministry of Environment of RA as a part of the Dilijan NP management plan for the 2007–2011 and 2017–2025 periods [6] p.215. High resolution satellite images (Pléiades) were procured within the UNDP-GEF "Mainstreaming sustainable land and forest management for the north-eastern mountain landscapes of Armenia" full size project (2016–2020) [12].

### 2.3. EO Data

As the analysis of land use and land cover and associated changes over the Dilijan national park spans from 1991 to 2019, covering 28 years, it was decided to use two satellite constellations: Landsat and Sentinel. As part of the Landsat family, Landsat 4, 5 and 7 times-series at 30 m spatial resolution were used for the mapping of the park in 1991, 1995, 2000, 2002, 2005 and 2010. Landsat 8 times-series at 30 m spatial resolution were used for 2015 and Sentinel-2 images at 10 m spatial resolution were used for the most recent mapping of 2019 (see Figure 2).

Two satellite VHSR data were used for calibration and validation purposes:

- Ikonos at 4 m spatial resolution for the year 2007, used for epoch 2005
- Pléiades of 17 July 2014 and 24 July 2014 at 0.5 m resolution (Panchromatic) and 2 m (Multispectral) used for epoch 2015 (see Figure 3)
- Pléiades of 28 June 2018 at 0.5 m resolution (Panchromatic) and 2 m (Multispectral), used for epoch 2019.

Data Pre-Processing

For Landsat data, the Level-2 Surface Reflectances have been processed using the ESPA on demand service. A total of 909 Landsat 4/5, 685 Landsat 7 and 323 Landsat 8 have been ordered. Using QA data (cloud detection), mosaics have been processed and cropped to the Dilijan National Park border.

For Sentinel data, the Level-2 flat Surface Reflectance using Maccs-Atcor Joint Algorithm (atmospheric correction and cloud screening software based on the MACCS processor, developed for CNES by CS-SI, from a method and a prototype developed at CESBIO) have been processed and downloaded from CNES Peps platform. After a resampling of the 20 m and 60 m bands to 10 m resolution, mosaics of Sentinel-2 imagery have been processed and cropped to the Dilijan National Park border.

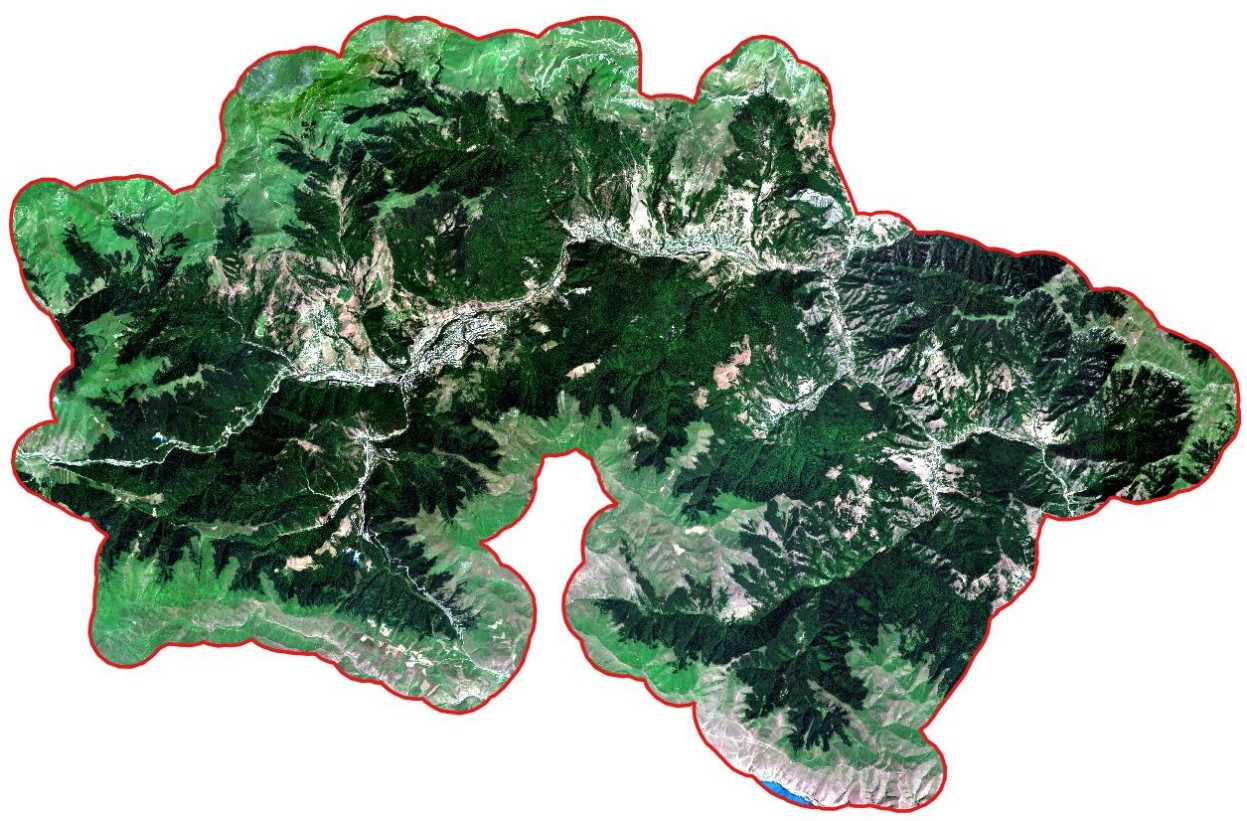

**Figure 2.** Sentinel-2 at 10 m spatial resolution acquired on 31 July 2019 @ESA.

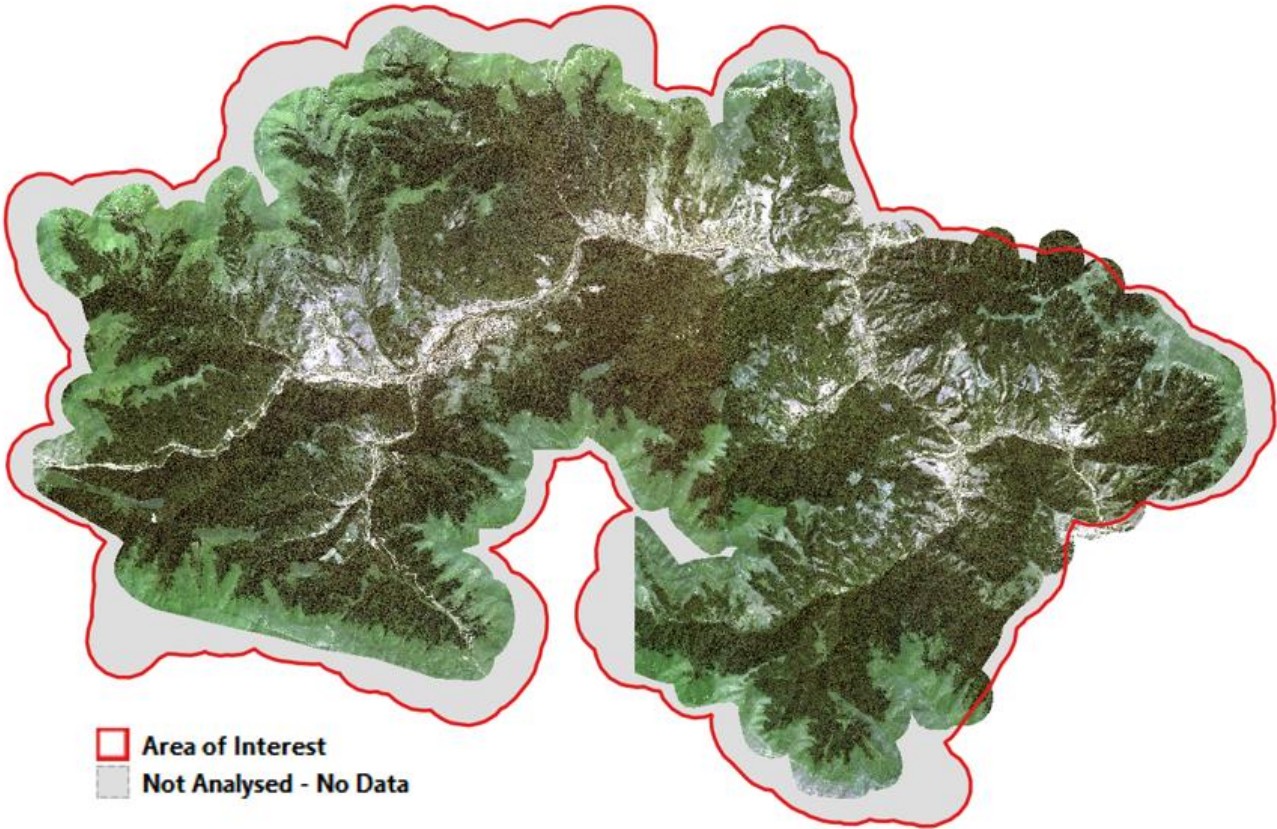

**Figure 3.** Pléiades at 0.5 m spatial resolution (Pansharpened) acquired on 17 and 24 July 2014 @CNES (2014), Distribution AIRBUS DS.

## 3. Methods

### *3.1. Characterization of Forest Ecosystems*

#### 3.1.1. Specifications

The Forest products consist of 9 Forest Density layers, 2 Forest Type layers and 7 Change layers representing the Forest Degradation/Deforestation. For the reference year 2019 where both Landsat and Sentinel-2 imagery were available, the status products are at both spatial resolutions of 30 m and 10 m. For consistency reasons, change products are only available at 30 m spatial resolution. The detailed Forest Products specifications are listed in Table 1.

**Table 1.** Product specifications for Service 2 Forest Mapping.

| General | |
|---|---|
| | **Forest Density**<br>2019—2 products available:<br><ul><li>10 m Sentinel-2</li><li>30 m Landsat 8</li></ul><br>2015—30 m Landsat 8<br>2010—30 m Landsat 5/7<br>2005—30 m Landsat 5/7<br>2002—30 m Landsat 7<br>2000—30 m Landsat 7<br>1995—30 m Landsat 5<br>1991—30 m Landsat 5 |
| Resolution and Data Input | **Forest Type & Dominant Leaf Type**<br>2019—30 m Landsat 8—Sentinel 2 |
| | **Forest Degradation/Deforestation**<br>Available for each subsequent epoch 1991–1995, 1995–2000, 2000–2002, 2002–2005, 2005–2010, 2010–2015, 2015–2019 as well as for the entire period (1991–2019) at 30 m |
| Geographic Projection | UTM Zone 38N |
| Format | GeoTIFF |
| Data Type | Byte |
| **Thematic information** | |
| | **Forest Density**<br><br>0–100: 0–100% Forest density (ground area covered by tree crowns)<br>255/NA–Outside of Area of Interest |
| | **Forest Type**<br><br>0: all non-forest areas<br>1: broadleaved forest<br>2: coniferous forest<br>255: outside area/no data |
| Classes and Coding | **Broadleaved and Coniferous Density**<br><br>0: Non-Forest<br>11: Pure broadleaved (>75%)<br>12: Dominantly broadleaved (50–75%)<br>21: Pure needle leaved (75%)<br>22: Dominantly needle leaved (50–75%) |
| | **Forest Degradation/Deforestation**<br><br>0: Non-forest stable<br>1: Forest regeneration<br>2: Deforestation<br>31: Anthropogenic forest degradation<br>32: Natural forest degradation<br>4: Forest stable |

<p style="text-align:center">**Table 1.** *Cont.*</p>

| Accuracies | |
|---|---|
| Geometric positional accuracy:<br>Thematic accuracy: | 1 pixel<br>85% |
| **Minimum Mapping Unit (MMU)** | |
| Sentinel-2 (10 m)<br>Landsat 5–8 (30 m) | 0.25 ha (25 px) for forested area (No MMU for other classes and for changes)<br>1 ha (11 px) for forested area (No MMU for other classes and for changes) |

### 3.1.2. Forest Mask

The Forest Mask is not a Forest Product, but it serves as a basis for the Land Cover/Land Use classification and associated changes, as well as for the Forest Density, the Forest Type and the Forest Degradation/Deforestation maps. The Forest Mask processing workflow can be divided into 6 main steps: (i) Pre-processing of EO data, (ii) Classification, (iii) Post-processing, (iv) Computing of the raw Forest Change Mask for all epochs, (v) Manual Enhancement of polygons of changes, (vi) Quality Check of the consistency of the Forest Masks and Forest Change Masks for all epochs.

(i)     Pre-processing

First, atmospheric, radiometric, and topographic corrections, using the MAJA algorithm for Sentinel-2 data, were applied on the time-series available, followed by a cloud masking. High Resolution Landsat and Sentinel-2 images for epoch n ± 1 in enlarged vegetation season from 1st of May to 31st of October were then selected in order to obtain a full mosaic coverage of the study area.

(ii)     Classification of the Forest Mask

Machine Learning methods based on the random forest algorithm is a common practice found in literature for the mapping of land use and land cover, and more specifically of forest areas [9,10].

The classification methodology is based on Geographic Object-Based Image Analysis (GEOBIA) using the Broceliande processing chain [13,14]. This methodology has proven its efficiency for the mapping of the Copernicus High Resolution Layers Small Woody Features and Forest for the reference years 2015 and 2018, at Pan-European scale covering an area of about 6 million km$^2$ [8,9]. The approach is actually a mixture of both GEOBIA and pixel-based analysis. It relies on two main components which are feature extraction based on attribute profiles, and a semi-supervised classification using a random forest algorithm at pixel level. The first step consists of computing predefined indexes such as the Normalized Difference Vegetation Index (NDVI), as well as texture characterization such as Sobel gradient and Haar-like features based on integral image representations. The Sobel gradient has proven to be more efficient than the renowned Haralick (1973) texture feature extraction based on a Gray Level Co-occurrence matrix (GLCM) which requires a time-consuming computation of a set of statistics for multiple distances and orientations. The texture information is extracted from each spectral band of the original EO multispectral image, as well as from the NDVI. This textural information will have a significant role in allowing the discrimination between Built-up and Bare soil, as their spectral response can be close. A first binary mask is produced, where all pixels flagged as no-data are set to a NoData value. Multi-scale features called Attribute Profiles (AP) and Differential Attribute Profiles (DAP) are derived through a model of morphological tree, which can be seen as a hierarchical segmentation: these trees allow to identify, depending on the way they are pruned, morphological objects from the EO data (see Figure 4). The second step is the use of the random forest classifier at pixel level.

Reference samples were collected, i.e., ~30 forest samples (examples) and 30 non-forest samples (counterexamples) to train the classifier. An in-house Machine Learning

object-oriented classification algorithm was used to derive a probability map, depicting the probability for each pixel to belong to the forest class. A manual thresholding approach enabled the conversion of the latter into a binary forest/non-forest layer.

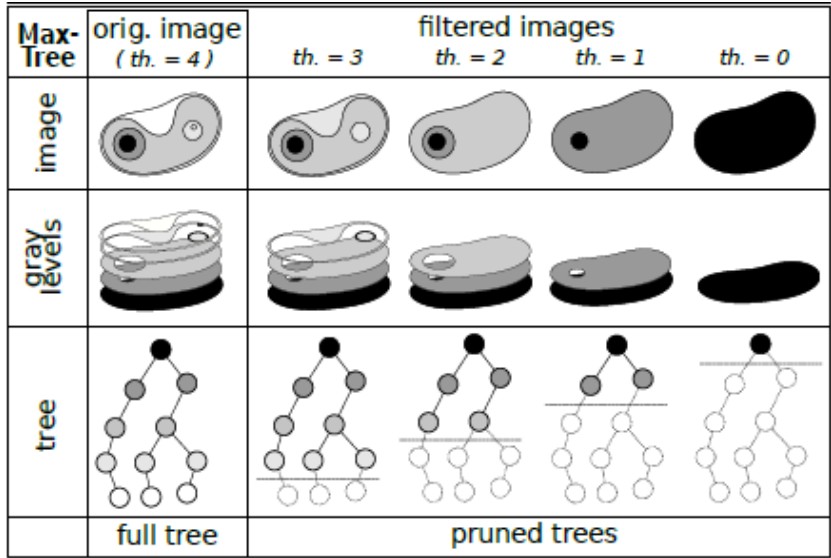

**Figure 4.** Principle of morphological trees derived from Attributes Profiles (example of Max-tree) to identify objects from the imagery.

(iii)   Post-processing

The post-processing step consisted of the application of a Minimum Mapping Unit (MMU) of 1 ha for Landsat data, i.e., 10 Landsat pixels at 30 m spatial resolution, and in a MMU of 0.5 ha for Sentinel-2 data, i.e., 25 Sentinel-2 pixels at 10 m spatial resolution.

(iv)   Computing of raw Forest Change Mask

The Forest Change Mask at 30 m resolution was then computed by subtracting the previous monitoring epoch from the most recent one, 2019–2015, 2015–2010, 2010–2005, 2005–2002, 2002–2000, 2000–1995, 1995–1991, as well as 2019–1991 over the whole 28-year monitoring cycle.

(v)   Manual Enhancement of polygons of change

The Forest Change Mask for each period was converted to vector format and each polygon $\geq$ 1 ha MMU was labelled as either real gain for the most recent epoch, omission from the previous epoch (undetected tree), commission from the most recent epoch (false tree), real loss for the most recent epoch, commission for the previous epoch (false tree), or omission from the most recent epoch (undetected tree). The polygons of changes were then rasterized, and the Forest Mask of each epoch recoded in accordance with the identified real or false changes. Finally, the Forest Mask 2019 was resampled from 30 m to 10 m, corresponding to the initial best available spatial resolution of Sentinel-2 data, and co-registered to the Forest Mask 1991 grid extent at 30 m spatial resolution.

(vi)   Quality Check of the consistency of the Forest Masks and Forest Change Masks for all epochs

Considering Forest Mask 2019 as the reference and the most precise product based on Sentinel-2 data at 10 m spatial resolution, the consistency of the changes between each epoch was checked retrospectively. Based on Forest Mask products generated for the 8 requested epochs (1991, 1995, 2000, 2002, 2005, 2010, 2015, 2019), forest changes were isolated and analyzed and served as a basis for the final forest degradation and deforestation map (See Figure 12).

### 3.1.3. Forest Density Mapping

The Forest Density is defined as the vertical projection of tree crowns to a horizontal earth's surface and provides information on the proportional tree canopy coverage per pixel, in the range of values from 1 to 100%. The method used for the estimation of Forest Density is similar to the one used for the Copernicus Land Monitoring Service High Resolution Forest at Pan-European level (https://land.copernicus.eu/pan-european/high-resolution-layers/forests, accessed 23 July 2021) and REDD+ (Reducing Emissions from Deforestation and Forest Degradation) Copernicus project, for example in African countries (https://www.reddcopernicus.info/, accessed 23 July 2021). Forest Density modelling is based on the multiple linear regression between reference samples and vegetation indexes derived from the HR times-series available. Forest Density reference data is assessed by means of visual interpretation of VHSR data following a point grid approach. The processing workflow of the Forest Density map can be divided into 4 main steps: (i) Sample drawing and interpretation, (ii) Computing of vegetation indexes, (iii) Multiple linear regression modelling, (iv) Quality Check of the consistency of the Forest Density between all epochs.

(i)    Sample drawing and interpretation

The first step consists in establishing a reference value of Forest Density for the sample units, based on the interpretation of VHSR imagery (Ikonos, Pléiades). The prerequisite is the co-registration of VHSR data to HR data, i.e., Ikonos 2007 and Landsat 5 nearest epoch 2005, Pléiades 2014 and Landsat 8 nearest epoch 2015, Pléiades 2018 and Sentinel-2 nearest epoch 2019. The sampling design is based on a stratified random sampling of 100 squared Primary Samples Units (PSU) of 100 × 100 m that intersect the forest stratum. Each PSU is then subdivided into 100 Secondary Sample Unit (SSU) of 10 × 10 m spacing. Each SSU is then visually interpreted as forest or non-forest. Zonal statistics are then computed for each PSU of 100 m$^2$ as the sum of SSUs interpreted as forest from which is then derived the percentage of forest density (see Figure 5 below).

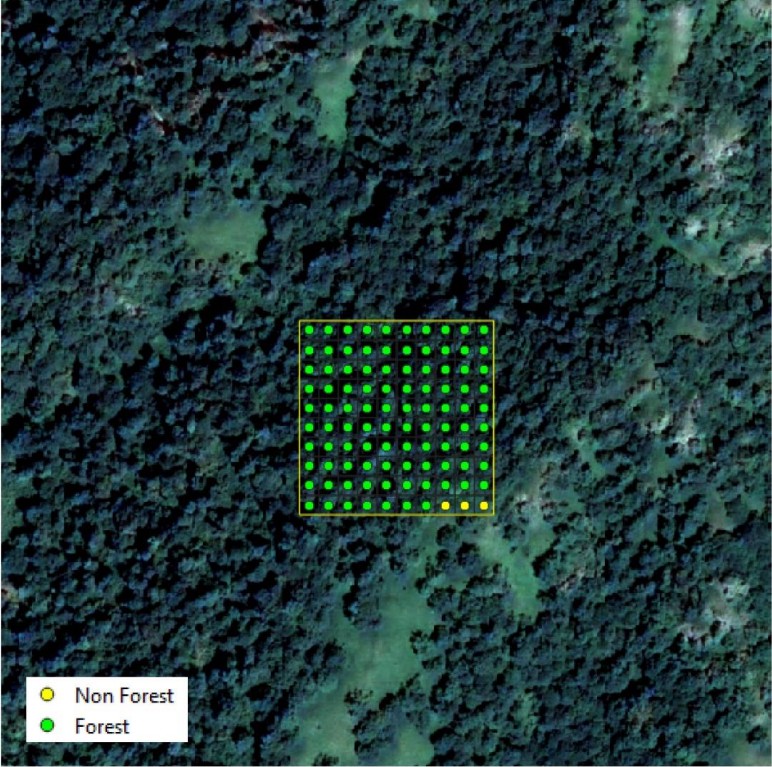

**Figure 5.** Visual interpretation of SSU points inside a PSU for computing the percentage of Forest Density reference value.

(ii)   Computing vegetation indices

The second step consists in computing several vegetation indexes within each PSU over the HR times-series, in order to establish a correlation between those indexes and the Forest Density value obtained through the interpretation of samples. Common indexes for the assessment of Forest and the characterization of vegetation and leaf properties are the Normalized Difference Vegetation Index (NDVI), Brightness Index (BI), Normalized Burnt Ratio (NBR), Green Normalized Difference Vegetation Index (GNDVI), Modified Normalized Difference Vegetation Index (GNDVI), Modified Normalized Difference Water Index (MNDWI), as well as the mean value of the spectral bands available, such as Blue, Green, Red, Near Infrared (NIR) and Short Wave Infrared (SWIR) [15].

Normalized Difference Vegetation Index (NDVI)

$$\text{NDVI} = \frac{\text{NIR} - \text{RED}}{\text{NIR} + \text{RED}} \tag{1}$$

Brightness Index (BI)

$$\text{BI} = \frac{(\text{RED} \times \text{RED})/(\text{GREEN} \times \text{GREEN})}{2} \tag{2}$$

Normalized Burn Ratio (NBR)

$$\text{NBR} = \frac{\text{NIR} - \text{SWIR}}{\text{NIR} + \text{SWIR}} \tag{3}$$

Green Normalized Difference Vegetation Index (GNDVI)

$$\text{GNDVI} = \frac{\text{NIR} - \text{GREEN}}{\text{NIR} + \text{GREEN}} \tag{4}$$

Modified Normalized Difference Water Index (MNDWI)

$$\text{MNDWI} = \frac{\text{RED} - \text{SWIR}}{\text{RED} + \text{SWIR}} \tag{5}$$

(iii)   Multiple linear regression analysis

The third step lies in defining the model that will predict the Forest Density value, through an analysis of the best multiple linear regression model, according to the following equation:

$$Y = a_1 x_1 + a_2 x_2 + a_3 x_3 + \ldots + a_n x_n + b \tag{6}$$

Y = estimated Forest Density value
$a_n$ = coefficients
$x_n$ = explanatory variables (vegetation indexes)
b = constant

Candidate models are assessed with the use of the Bravais-Pearson correlation coefficient, and the model with the higher coefficient was selected.

$$r = \frac{\frac{1}{n} \sum_{i=1}^{n} (x_i - \bar{x})(y_i - \bar{y})}{\sigma_x \times \sigma_y} \tag{7}$$

$\bar{x}$ = mean
$\sigma_x$ = standard deviation

Finally, the most accurate model was applied over the complete area of interest in order to derive the Forest Density values. An example of Forest Density for the most recent 2019 epoch is shown in Figure 8.

(iv)   Quality Check of the consistency of the Forest Density between all epochs

The Computed Forest Density value over a stable area can present change over 1991–2019 due to several factors: for instance, EO imagery from various sensors or acquisition time of EO image reflecting seasonality. To avoid artificial changes in Forest Density value, a smoothing of the Forest Density was applied in order to ensure the consistency between the different epochs. Small variations of values between epochs were then considered as artefacts unless visually interpreted as real changes on EO data. For those stable areas, Forest Density values were then averaged between two successive epochs.

### 3.1.4. Forest Types

According to the referenced literature [3], Dilijan NP mainly consists of deciduous species such as oak (*Quercus iberica*, *Q. macranthera*), oriental beech (*Fagus orientalis*), common and oriental hornbeam (*Carpinus betulus*, *C. orientalis*), which form homogeneous oak, beech and hornbeam forests as well as mixed forests with different combinations of the species mentioned. Coniferous forests (pine–*Pinus*, juniper–*Juniperus* and yew–*Taxus*) occupy a limited territory in the national park and occur in patches. Pine often makes dense forests in the basin of the River Hovajur on the slopes of the Areguni and Pambak ranges in the vicinity of serpentine Dilijan highway. There are many pine trees in Dilijan and on nearby slopes. Additionally, there are no *Larix sp.*, neither *Quercus Sempervirens* at all in Armenia. *Buxus* is grown only artificially for greening purposes in settlements.

The Forest Type map is the distinction between broadleaved and coniferous forest. Broadleaved Density and Coniferous Density completes this map as it provides the density of broadleaved and coniferous forest at 10 m spatial resolution for the 2019 epoch. To derive those maps, the processing workflow is divided into 3 steps: (i) Identification of coniferous reference samples and analysis of their spectral signature over the EO data time-series, (ii) Classification over the whole area, (iii) Crossing with Forest Density to derive Broadleaved Density and Coniferous Density.

(i)     Identification of coniferous reference samples and analysis of their spectral signature over the EO data times-series

Using UNDP ancillary field data provided in the frame of this study, coniferous parcels could be identified in 2007. Visual interpretation over the 2019 Sentinel-2 scenes, using these 2007 parcels as reference, could allow the determination of training samples of both broadleaved forest area and coniferous forest area inside the 2019 forested area. NDVI was computed for 5 Sentinel-2 scenes over the 4 annual seasons, to establish "NDVI signatures" of both tree types (see Figure 6 below).

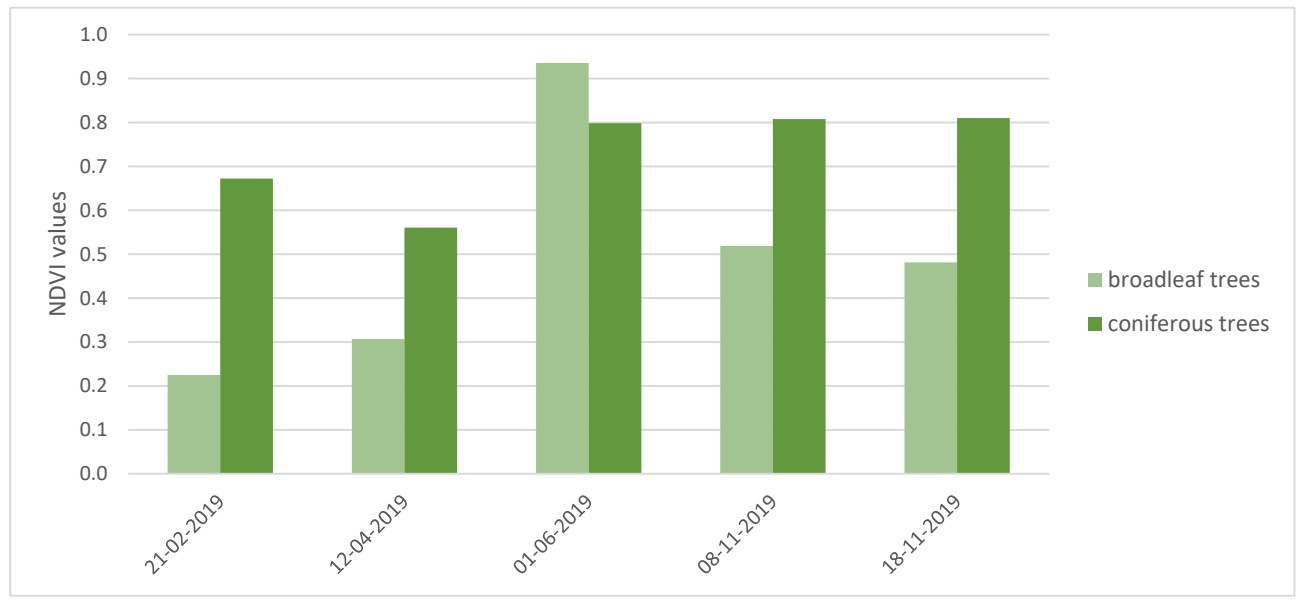

**Figure 6.** Coniferous and broadleaves samples NDVI signature over 5 periods of the year.

(ii)   Classification over the whole area

Using the NDVI signatures derived in the previous section, it was possible to derive, inside the 2019 forest mask, the discrimination between broadleaved and coniferous trees. Using several observations allowed the reduction in artefacts that could derive from shadows.

Given the very limited area covered by coniferous species, the quality of the Forest Type map was assessed by means of visual interpretation over reference field data provided by UNDP Armenia.

(iii)   Crossing with Forest Density to derive Broadleaved and Coniferous Density

Once the coniferous/broadleaf layer was produced, it was possible to combine it with the Forest Density 2019 product to derive the Broadleaved and Coniferous Density products, with the following nomenclature:

- Needle leaved trees

    - Pure needle leaved (75%)
    - Dominantly needle leaved (50–75%)

- Broadleaved trees

    - Pure broadleaved (>75%)
    - Dominantly broadleaved (50–75%)

*3.2. Land Use and Land Cover Classification and Associated Changes*

3.2.1. Specifications

To support land use planning in the Dilijan National Park, the production of standardized land use and land cover (LULC) maps that show the current baseline and highlight past changes and trends was necessary. The production was divided into eight epochs–for each of these epochs a dedicated map was produced. Table 2 lists all of these epochs as well as the EO input data used for production.

**Table 2.** Production epochs for the LULC maps and input satellite data.

| Epoch | Satellite Data | Resolution | Comment |
|-------|---------------|------------|---------|
| 2019 | Sentinel-2 | 10 m | Most recent situation |
| 2015 | Landsat 8 | 30 m | |
| 2010 | Landsat 5 | 30 m | Use of Landsat 5 (due to Landsat 7 SLC error) |
| 2005 | Landsat 5 | 30 m | Use of Landsat 5 (due to Landsat 7 SLC error) |
| 2002 | Landsat 7 | 30 m | Status change from state reserve to national park |
| 2000 | Landsat 7 | 30 m | |
| 1995 | Landsat 5 | 30 m | |
| 1991 | Landsat 5 | 30 m | Armenian Independence |

The target resolution of the final maps corresponds to the resolution of the EO data that was used as an input for the production of each year. However, for the most recent year (2019) two separate thematic maps were produced: Firstly, a map in 30 m to achieve better comparability to previous epochs and secondly, a 10 m map to facilitate a potential future Sentinel-2 based monitoring system of the national park, which would require a 10 m thematic map as a baseline. Furthermore, the minimum mapping unit (MMU) of objects on the ground was chosen based on the production year as well. The year 2019 was specified to have an MMU of 0.25 ha and all other years (based on Landsat) have an MMU of 1 ha.

Finally, the thematic content of the maps was specified in accordance with the stakeholders and covers the following classes:

1.   Forest
2.   Agriculture (arable land and pastureland)

3. Settlements
4. Primary roads
5. Bare soil
6. Other vegetated areas
7. Water bodies
8. Rivers

3.2.2. Land Use and Land Cover Classification Method

The LULC classification workflow can be divided into several parts: (i) Pre-processing, (ii) Feature Extraction, (iii) Training, (iv) Classification and (v) Post-processing. These steps are subsequently described in more detail below:

(i)　Pre-processing

The pre-processing includes the cloud masking of all EO input data and a reprojection into the same geographic grid and projection UTM 38N. For Landsat data, the cloud masking is performed by using the provided quality layer (BQA) which provides information in clouds, cloud shadow, cirrus, snow and ice and cloud confidence. For Sentinel-2 the MAJA cloud masks were used to mask out saturated or defected pixels, dark areas, snow, thin cirrus clouds as well as clouds with high and medium probability and their shadows.

(ii)　Feature Extraction

The feature extraction first includes the derivation of meaningful indexes that will help distinguish vegetated and non-vegetated classes. For this reason, for each available image, the Normalized Difference Vegetation Index (NDVI) as well as the Normalized Difference Water Index (NDWI) was calculated. After this step of feature engineering, the data were temporarily aggregated: For each spectral band and year, three percentiles (p) were derived: p20, p50 (median) and p80. This temporal aggregation is used to reduce the amount of data for the model and therefore decrease training and processing time as well as for the purpose of creating a less complex model (with less input features) with the capabilities of better generalization.

(iii)　Training

The training of every supervised classification task requires reference data. These are already to some extent described in Section 2.2. However, two key datasets that were essential to the LULC classification are described here in more detail. This on one hand includes the Global Crop Extent Layer which was produced by Pittman et al. (2010) [16] and serves as the only available input dataset on agricultural activities in the area. Secondly, the Open Street map dataset which offers a very high accuracy and level of completion in the area of interest and serves as training data for most thematic classes: Settlements, Roads, Water bodies and rivers. The other classes (Bare Soil and other vegetation) were sampled manually with the help of very high-resolution data as well times-series information from Sentinel-2 imagery.

(iv)　Classification

The classification itself was performed with the highly popular tree based Random Forest classifier [17], the training parameters (although often neglected in case of random forest) were tuned via cross validation and overall accuracy as a target metric. The final model with 500 trees yielded an out-of-bag (OOB) error of 14.6% for the classification of the most recent year 2019 based on Sentinel-2 data.

(v)　Post-Processing

The subsequent post-processing included an overlay and insertion of thematic information from Open Street Map and Service 2 with the aim of enhancing the accuracy of the product. Overall, it was decided to take over the following thematic classes from Open Street Map: Settlements, roads, water bodies and rivers. This was done for the reason that a human-based manual very detailed delineation of these object was judged to be superior

to the artificial intelligence-based classification based on high resolution imagery. The Forest class was entirely taken over from Service 2. Furthermore, the MMU was applied to the classification, however it has to be stressed that for Settlements and Primary Roads the MMU was lowered to 0.02 ha. This step was necessary to preserve the small-scale scattered structure of these anthropogenic objects in the area. A larger MMU would yield an underestimation of these areas.

Please note that the above elaborated workflow only describes the classification task for the baseline epoch 2019. All other epochs and their maps were produced with a subsequent change detection approach that compares the spectral characteristics of each year to the baseline year, flags noticeable changes in the area and subsequently reclassifies them into the correct class. This procedure is described in more detail in the next chapter.

### 3.2.3. Change Detection

To assess historical land cover changes, a spectral change vector analysis can be performed if the sensor and band properties are consistent over time (i.e., the Landsat Legacy). Using this method, spectral properties of the reference year 2019 can be compared to its historic spectral properties for each epoch and for each pixel, respectively. If a spectral difference between two compared epochs for a certain pixel lies above a predefined threshold, a potential change is identified and flagged.

A high change vector length value is, for example, expected during urbanization activities (i.e., the conversion of vegetation into sealed surface). Low change vector lengths may occur if there is no change at all and the spectral properties only differ due to meteorological effects, i.e., because of a delayed or shortened rain season or also during vegetation regrowth due to afforestation activities.

The applied change vector analysis is not based on a single comparison of the spectral properties of two spectral bands but considers the comparison of the entire spectrum and can therefore be called a multidimensional change vector analysis.

As already highlighted above, the algorithm considers each epoch of the reference period (1991–2019) and compares it to the reference year (2019). Ultimately, change areas are detected by dynamically thresholding each comparison individually via a high percentile value (i.e., 0.99).

Figure 7 depicts this approach in a two-dimensional space where small change vector lengths are considered stable and high differences are flagged as change. By summing up the occurrences of abnormally high differences over time per pixel, a probability of change or even an exact timing of change can be derived if of interest.

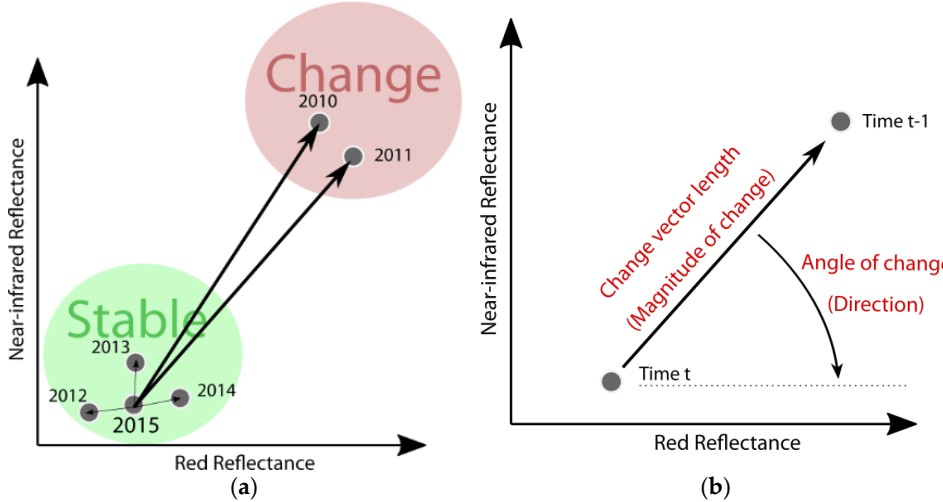

**Figure 7.** (**a**) left: Example of a two-dimensional change vector analysis between two timesteps using the red and near-infrared spectral bands and (**b**) right: Example of a time-series-based two-dimensional change vector analysis ©GeoVille.

The identified "change candidate areas" are then flagged and reclassified into target change classes via a supervised classification. The algorithm itself is a supervised stratified random forest classifier as described above.

### 3.2.4. Product Validation

The validation of the final land cover maps is based on the good practices described by Olofsson et al. (2013) and Congalton et al. (1991) [18,19]. Therefore, the sampling design was chosen in a way that each class sample size is large enough to produce sufficiently precise estimates of the area of the class, for that reason the minimum sample size per stratum (class) was set to 30. The equation below calculates an adequate overall sample size for stratified random sampling that can then be distributed among the different strata.

$$n = \frac{\left(\sum W_i S_i\right)^2}{\left[\hat{S}(O)\right]^2 + \frac{1}{N} \sum W_i S_i^2} \approx \left(\frac{\sum W_i S_i}{\hat{S}(O)}\right) \tag{8}$$

N is the number of units in the area of interest (number of overall pixels)
S(O) is the standard error of the estimated overall accuracy (set to 0.05)
$W_i$ is the mapped proportion of area of class
$S_i$ is the standard deviation of stratum i

## 4. Results and Discussion

### 4.1. Forest Ecosystems Characterization

#### 4.1.1. Distribution of Forest Densities and Their Evolution from 1991 to 2019

An illustration of a Forest Density map is given in Figure 8 below for the last epoch 2019. Dilijan National Park is composed mainly of dense mountainous forest >75%. Despite the harmonization of Forest Density values between different epochs explained in the methodology Section 3.1.3, the latter are strongly correlated to the specific multiple regression linear model used for each epoch and the corresponding EO imagery used to calculate these models. At the beginning of the monitoring cycle, in the 1990s, a strong forest density loss reveals uncontrolled logging and increasing anthropogenic pressure on the natural landscape. The middle period right after the creation of the Park 2002–2005 shows almost a complete stop of clear-cuts and the overall picture is a stable area, corresponding to protective measures taken by the Armenian Government. Real losses in forest density changes observed for the latest 2015–2019 period account for a rebound of forest degradation and deforestation. Despite this threat, and although the aim of this study did not include reforestation or afforestation, it is an encouraging sign to observe that there is also a recent increase in forest density which coincides with natural forest regeneration all across Dilijan National Park.

The validation of the Forest Density products comes into the form of a correlation between reference values (interpretation of samples on VHSR EO data) and modelized Forest Density values present in the corresponding product, as show in Figure 9 below.

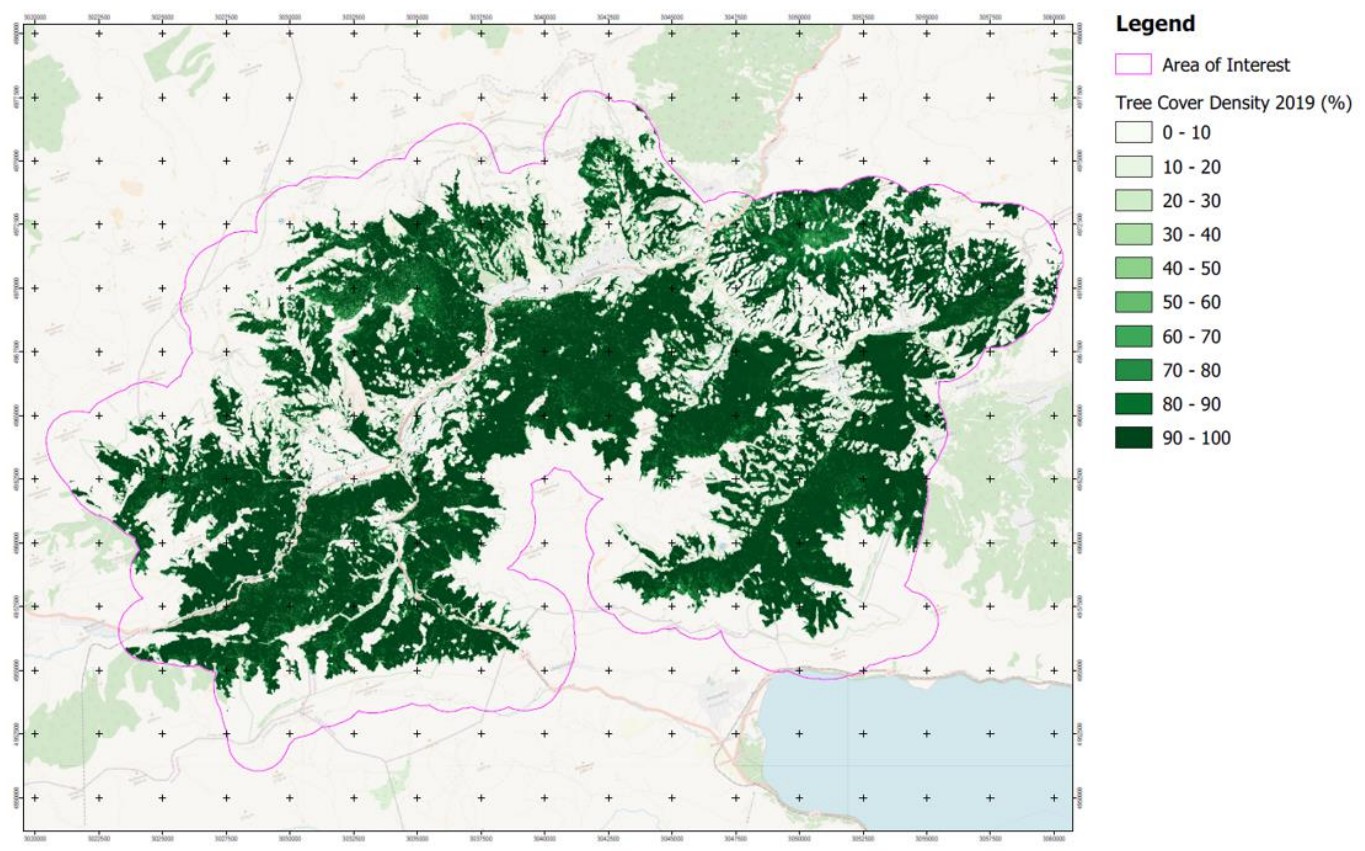

**Figure 8.** Forest Density year 2019.

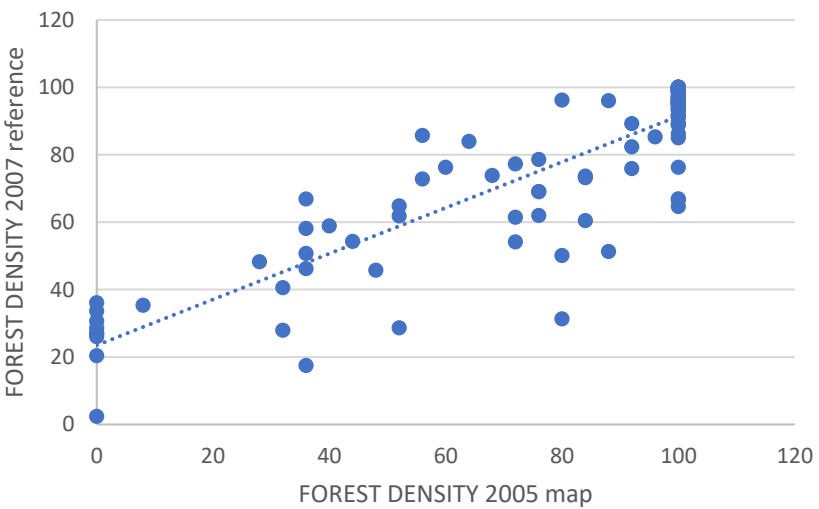

**Figure 9.** Correlation between Forest Density 2007 reference (computed from interpretation of SSUs) and Forest Density 2005 map values ($R^2 = 0.7756$).

The correlation coefficient between reference and map Forest Density values for each epoch is presented in Table 3 below.

**Table 3.** Correlation between Forest Density value of reference data and map product.

| Epoch | Coeff. Correlation R² between Reference and Product |
|-------|---------------------------------------------------|
| 1991 | 0.7559 |
| 1995 | 0.6988 |
| 2000 | 0.6991 |
| 2002 | 0.7864 |
| 2005 | 0.7903 |
| 2010 | 0.7988 |
| 2015 | 0.8336 |
| 2019 | 0.8901 |

### 4.1.2. Characterization of Forest Types

Analysis of 2019 EO data times-series allowed stating that Dilijan National Park forests are mainly constituted of broadleaved tree species (98.85% of Forested areas) and that coniferous species only represent a minor part of the forest included in this park (1.15% of forested areas) (see Figure 10 below).

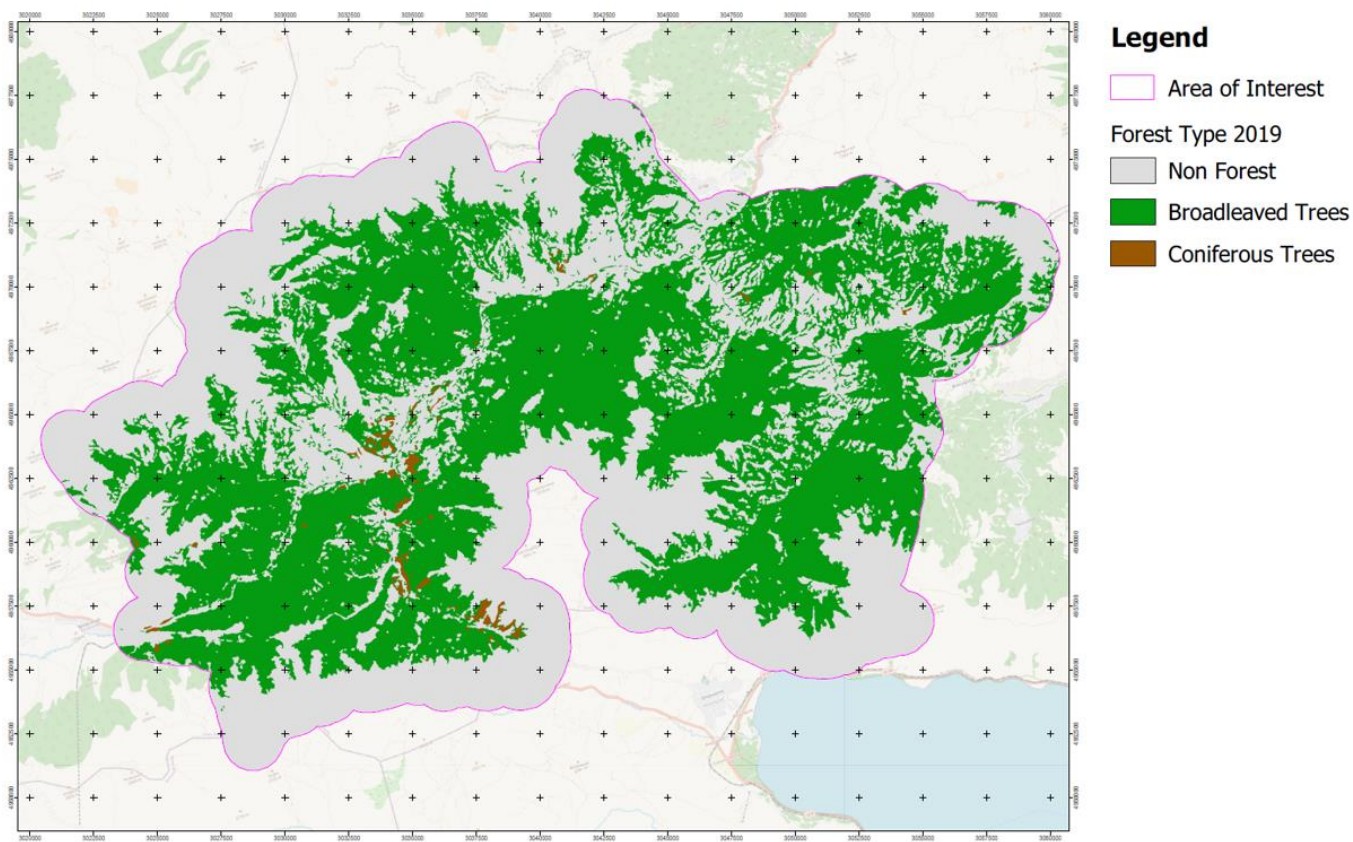

**Figure 10.** Forest Type year 2019.

### 4.1.3. Characterization of Deforestation and Forest Degradation between 1991 and 2019

Results concerning the forest perturbation (e.g., deforestation, degradation, or refor-estation) over the study period can be found in Table 4 and Figures 11 and 12 below.

**Table 4.** Forest perturbation on Dilijan National Park over the 1991–2019 period.

|  | **1991–1995** | **1995–2000** | **2000–2002** | **2002–2005** | **2005–2010** | **2010–2015** | **2015–2019** |
|---|---|---|---|---|---|---|---|
| Forest regeneration (ha) | 26 | 25 | 24 | 47 | 178 | 44 | 104 |
| Forest loss (ha) | 253 | 47 | 41 | 0 | 149 | 160 | 45 |
| Forest degradation (anthropogenic) (ha) | 384 | 363 | 12 | 22 | 72 | 46 | 53 |
| Forest degradation (natural) (ha) | 0 | 0 | 0 | 0 | 1 | 5 | 0 |

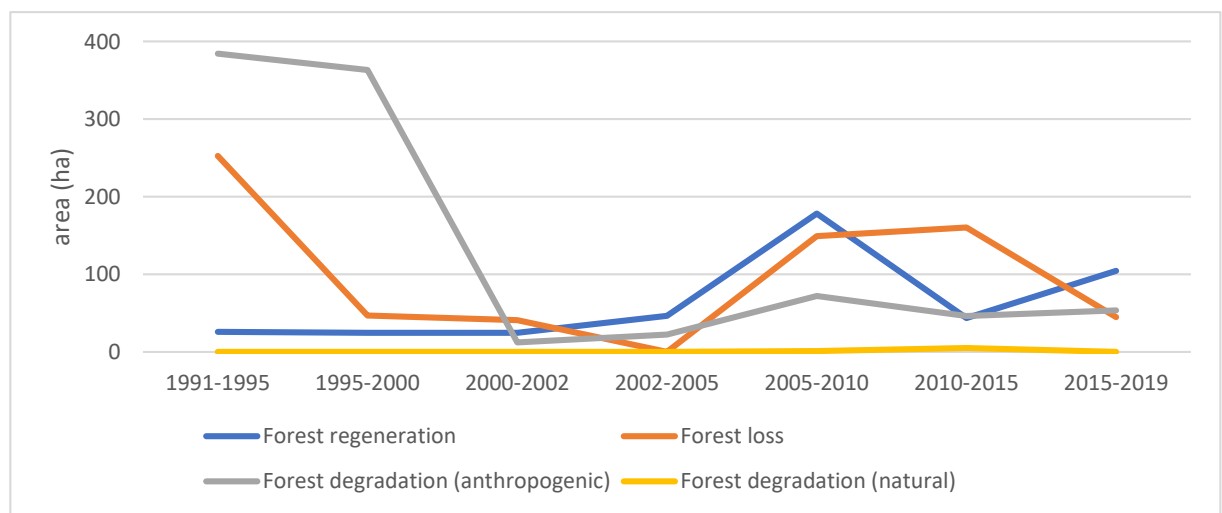

**Figure 11.** Evolution of Forest Perturbation in hectares, over the 1991 to 2019 period.

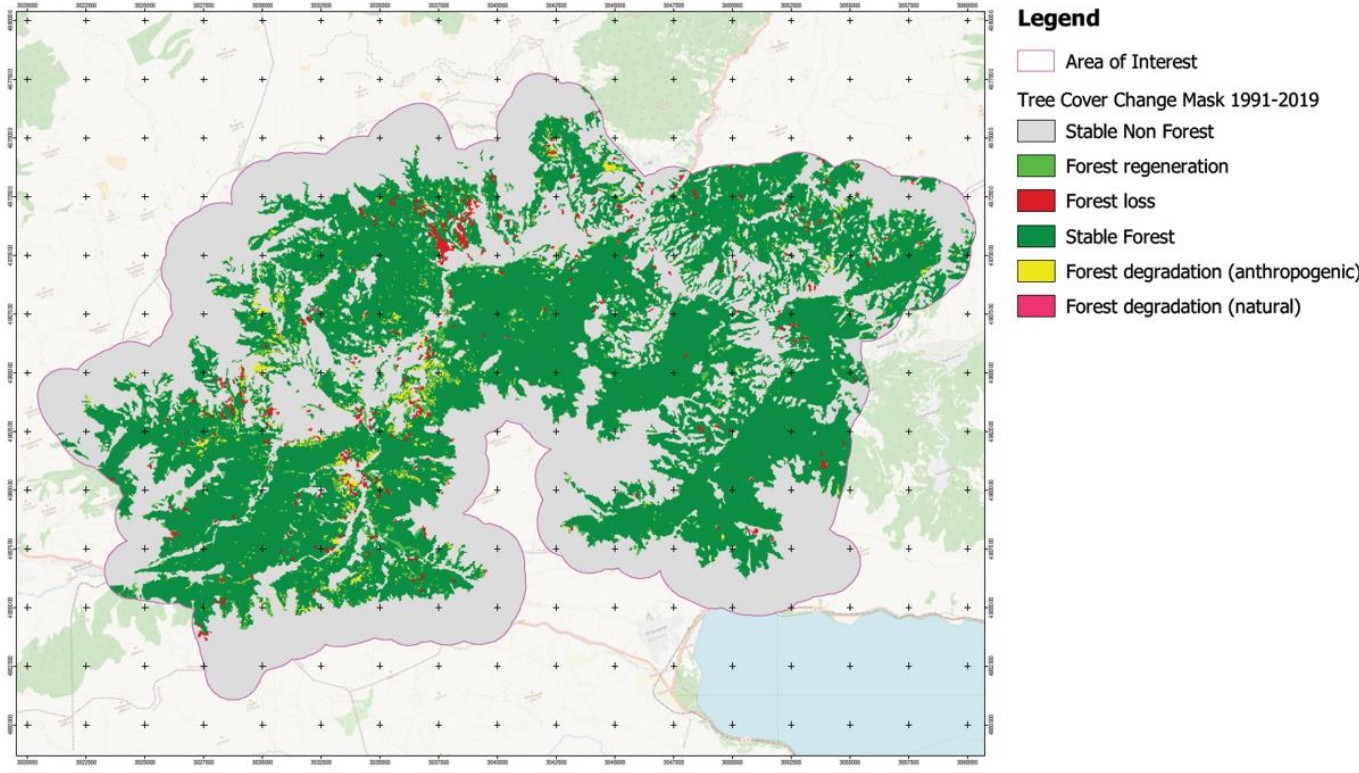

**Figure 12.** Forest disturbance map over the period 1991 to 2019.

With the creation of the National Park in 2002, the forest degradation drastically dropped, and the forest regeneration increased, following the set-up of forestry management policies.

The forest degradation, which is stable over the 2002–2019 period, has been attributed to anthropogenic origin unless ancillary data can prove that the cause is natural. Therefore, it is very likely that the anthropogenic forest degradation is strongly overestimated: forest wildfire monitoring service (NASA FIRMS) only monitors wildfires since year 2000, and no field data regarding other natural causes (storm, disease, etc.) were available.

### 4.2. Land Use and Land Cover Change

The result of the LULC classifications as well as the mapped changes are subsequently presented below. This includes the presentation of the final LULC maps, the comparison of the 10 m and 30 m products, the highlighting of areas with prominent changes as well as the presentation of derived land cover class areas for each year.

#### 4.2.1. Main Land Use and Land Cover Types

The results from the 2019 baseline LULC mapping based on Sentinel-2 (10 m) can be seen in Figure 13. In total, eight different classes were mapped, and the map shows that the area is dominated by forested areas with agricultural activities around several smaller settlements which function as a pressure regarding deforestation.

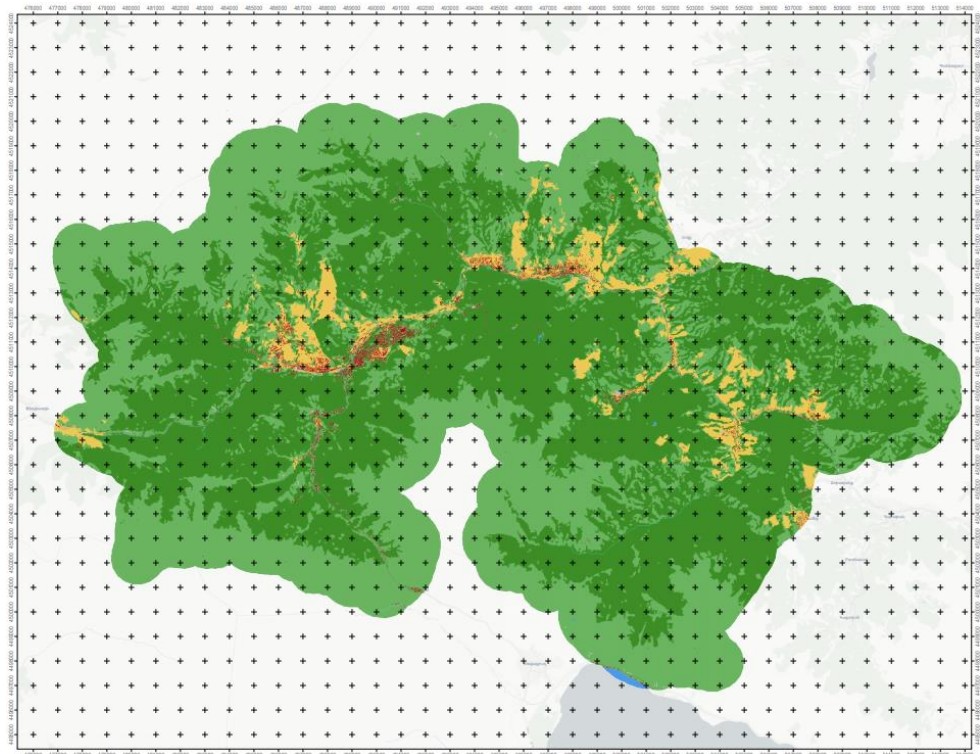

**Figure 13.** Dilijan National Park, Armenia: Land Cover Land Use Map 2019 based on Sentinel-2 data (10 m).

The total area breakdown is presented in Figure 14 and shows that natural vegetation makes up roughly 93.7% in the area of interest, with more than half of that area covered by forest. The remaining part of the area is mostly anthropogenically altered or used to some extent: 4.62% is dedicated to agriculture, 1.19% of the area is sealed (settlements and primary roads) and 0.46% are made up of standing water and rivers.

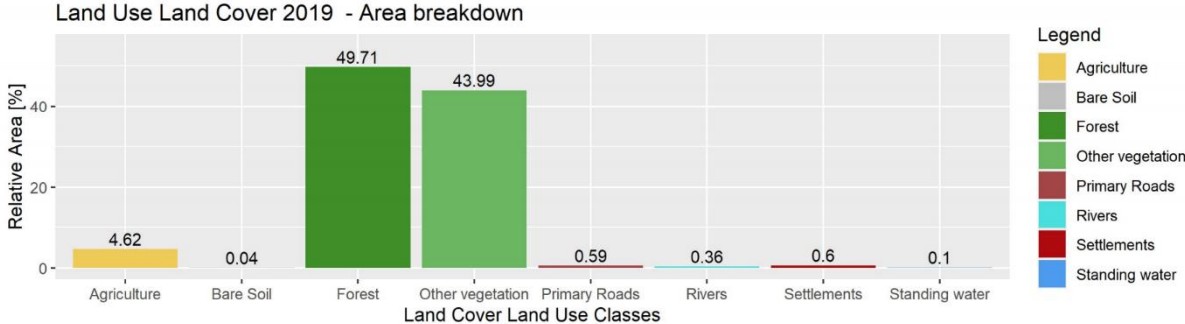

**Figure 14.** Breakdown of the area of land cover land use classes for the 2019 epoch.

As already mentioned above in the methodology section, two separate LULC maps for the year 2019 were produced at 10 m and 30 m resolutions from Sentinel-2 and Landsat 8 imagery, respectively. The Sentinel-2-based map was created to include the best available data (in the case of Sentinel-2) to produce a high-resolution map as a baseline for potential future monitoring activities. The Landsat 8 map was generated to achieve a better comparability between the preceding epochs, which are all based on Landsat 8 (because the first Sentinel-2 satellite was launched in 2015). Figure 15 shows both classifications side by side highlighting the area around the city of Dilijan inside the national park. On the left side, the Sentinel-2-based product at 10 m is shown and when directly compared to the Landsat 8 (right side) at 30 m resolution, it can be clearly seen that Sentinel-2 offers much greater spatial detail. On the left side even small (groups of) buildings are visible and due to the smaller minimum mapping unit, a more fragmented and detailed overview of the area is discernable.

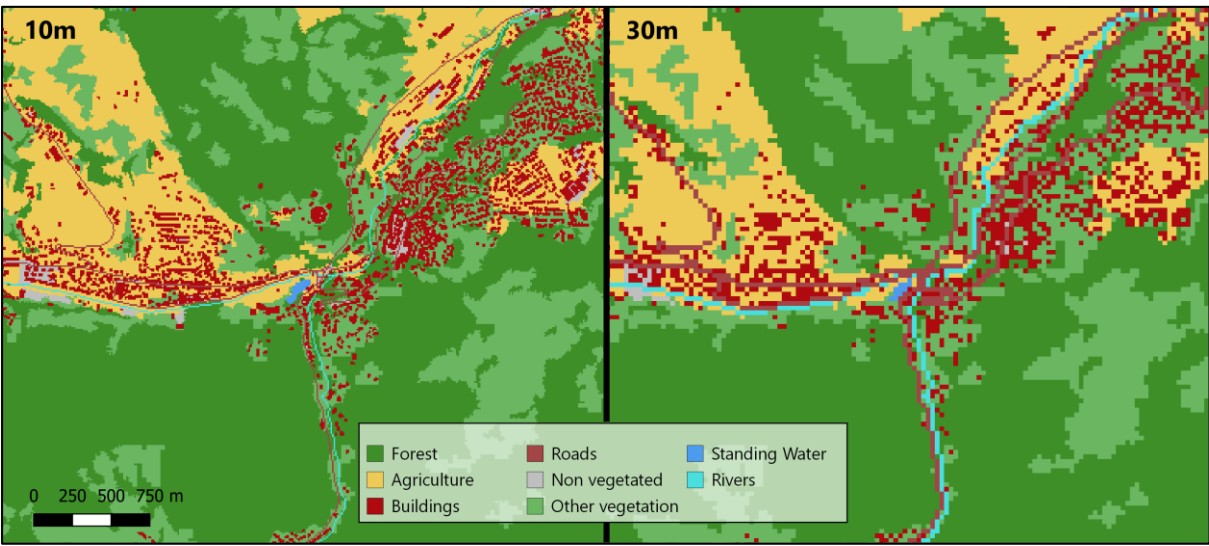

**Figure 15.** Comparison of the 10 and 30 m LULC map product for 2019. The Sentinel-2-based (10 m) product on the left and the Landsat 8-based product (30 m) on the right.

### 4.2.2. Mapping of Changes from 1991 to 2019

This subsection presents the results of the LULC maps for the epochs 1991–2019. Since the area of interest is mostly inside the Dilijan National Park where conversation practices are put in place, the area is very stable in terms of land cover change. Figure 16 depicts the area breakdown for four selected epochs (1991, 2002, 2010 and 2019) and clearly shows that most class areas remained almost the same over the entire period of time. The biggest change can be observed in the forest class that dropped from 50.18% in 1991 to 49.71% in

2019. The main driver behind this loss of forest is either natural degradation, small scale urbanization or minor agricultural expansion.

The class "other vegetation" (than forest) increased by 0.41% over the entire period and is the main driver behind the loss in forest, this is followed by a minor increase in agricultural activity of 0.04% and lastly by the extension of settlements which grew by 0.01% of the total project area from 1991 until 2019. All other classes (water and bare soil) did not show any significant change.

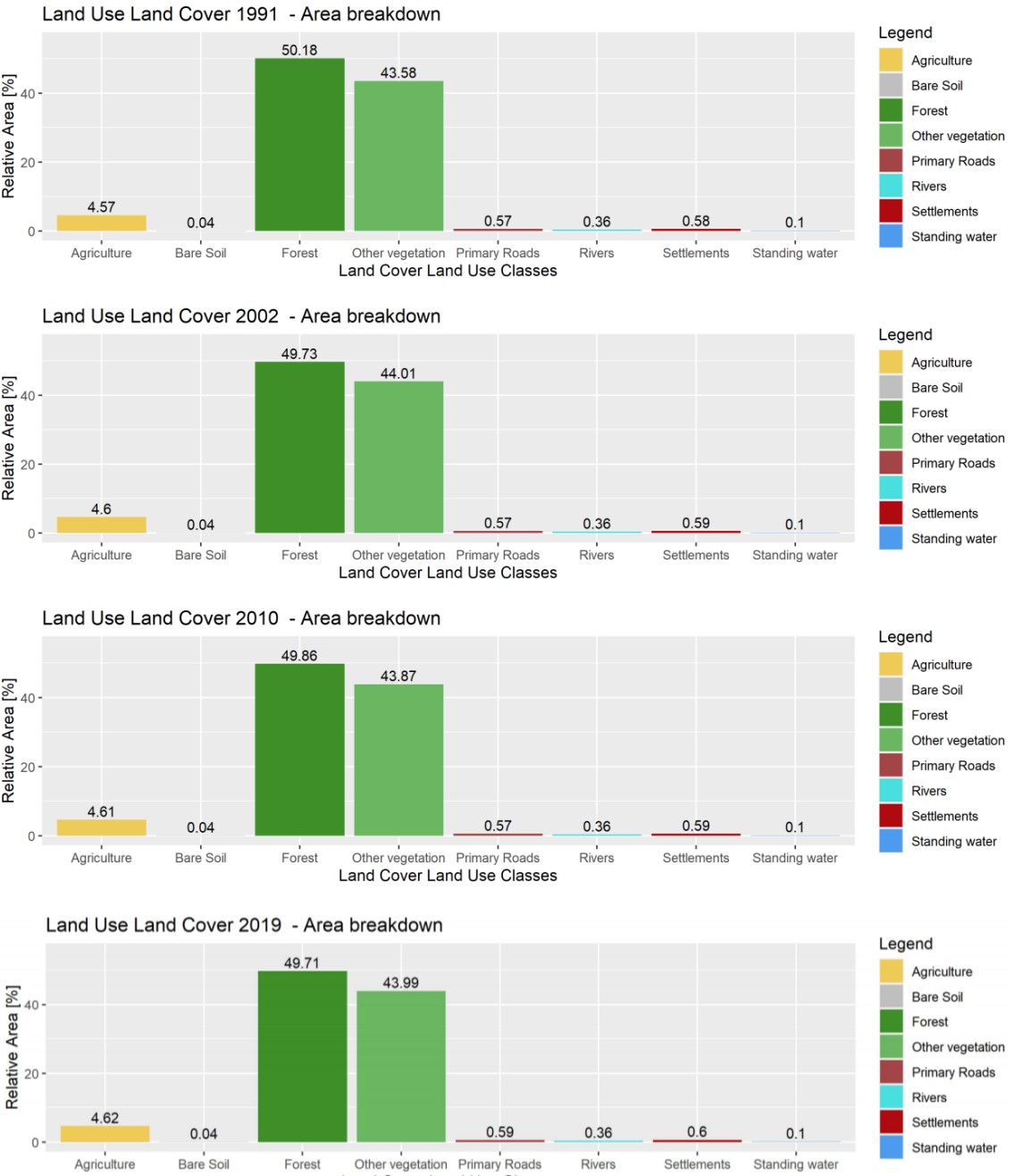

**Figure 16.** Breakdown of the area of land cover land use classes for the epochs 1991, 2002, 2010 and 2019.

The validation results for epochs 2019, 2002 and 1991 are presented in Tables 5–7 below and range between 85 and 89% in terms of overall accuracy. Generally speaking, the selected method together with the input data yields good results and most classes show both high user and producers' accuracies (usually above 80%). The class with the lowest

accuracies is "Other vegetation" whose producers' accuracy only ranges from 58%-63%. When looking at the validation matrices, it can be seen that this class often gets confused with agriculture. This probably arises from the fact that the training data for agriculture do not stem from a very reliable source (Global Crop Extent Layer) and the very broad class definition "Other vegetation" seems to be too vague for the classifier to be able to find sharp class boundaries which yields high confusion with other classes.

**Table 5.** Validation Matrix for Service 1: LULC classification for epoch 1991.

| | | Ground Truth | | | | | | | | # Samples | User Accuracy |
|---|---|---|---|---|---|---|---|---|---|---|---|
| | | Forest | Agriculture | Settlement | Primary Roads | Bare Soil | Other Vegetation | Standing Water | Rivers | | |
| Classification | Forest | 26 | 0 | 0 | 0 | 0 | 4 | 0 | 0 | 30 | 86.67% |
| | Agriculture | 1 | 22 | 0 | 0 | 0 | 7 | 0 | 0 | 30 | 73.33% |
| | Settlement | 0 | 1 | 23 | 1 | 2 | 3 | 0 | 0 | 30 | 76.67% |
| | Primary roads | 2 | 0 | 0 | 28 | 0 | 0 | 0 | 0 | 30 | 93.33% |
| | Bare soil | 0 | 0 | 0 | 2 | 23 | 5 | 0 | 0 | 30 | 76.67% |
| | Other vegetation | 1 | 1 | 0 | 0 | 0 | 28 | 0 | 0 | 30 | 93.33% |
| | Standing water | 1 | 0 | 0 | 0 | 0 | 0 | 29 | 0 | 30 | 96.67% |
| | Rivers | 1 | 0 | 0 | 0 | 1 | 1 | 0 | 27 | 30 | 90.00% |
| | Totals | 32 | 24 | 23 | 31 | 26 | 48 | 29 | 27 | **240** | |
| | Producer accuracy | 81.25% | 91.67% | 100.00% | 90.32% | 88.46% | 58.33% | 100.00% | 100.00% | | |
| | Overall accuracy | 85.83% | | | | | | | | | |

**Table 6.** Validation Matrix for Service 1: LULC classification for epoch 2002.

| | | Ground Truth | | | | | | | | # Samples | User Accuracy |
|---|---|---|---|---|---|---|---|---|---|---|---|
| | | Forest | Agriculture | Settlement | Primary Roads | Bare Soil | Other Vegetation | Standing Water | Rivers | | |
| Classification | Forest | 26 | 0 | 0 | 0 | 0 | 4 | 0 | 0 | 30 | 86.67% |
| | Agriculture | 0 | 24 | 0 | 0 | 0 | 6 | 0 | 0 | 30 | 80.00% |
| | Settlement | 0 | 0 | 28 | 1 | 0 | 1 | 0 | 0 | 30 | 93.33% |
| | Primary roads | 2 | 0 | 0 | 26 | 0 | 0 | 0 | 0 | 28 | 92.86% |
| | Bare soil | 0 | 0 | 0 | 5 | 22 | 3 | 0 | 0 | 30 | 73.33% |
| | Other vegetation | 3 | 1 | 0 | 0 | 0 | 26 | 0 | 0 | 30 | 86.97% |
| | Standing water | 1 | 0 | 0 | 0 | 0 | 0 | 28 | 1 | 30 | 93.33% |
| | Rivers | 1 | 0 | 0 | 0 | 0 | 1 | 0 | 28 | 30 | 93.33% |
| | Totals | 33 | 25 | 28 | 32 | 22 | 41 | 28 | 29 | **238** | |
| | Producer accuracy | 78.79% | 96.00% | 100.00% | 81.25% | 100.00% | 63.41% | 100.00% | 96.55% | | |
| | Overall accuracy | 87.39% | | | | | | | | | |

**Table 7.** Validation Matrix for Service 1: LULC classification for epoch 2019.

| | | Ground Truth | | | | | | | | # Samples | User Accuracy |
|---|---|---|---|---|---|---|---|---|---|---|---|
| | | Forest | Agriculture | Settlement | Primary Roads | Bare Soil | Other Vegetation | Standing Water | Rivers | | |
| Classification | Forest | 29 | 0 | 0 | 0 | 0 | 1 | 0 | 0 | 30 | 96.67% |
| | Agriculture | 1 | 22 | 0 | 0 | 0 | 7 | 0 | 0 | 30 | 73.33% |
| | Settlement | 1 | 0 | 26 | 1 | 1 | 1 | 0 | 0 | 30 | 86.67% |
| | Primary roads | 1 | 0 | 0 | 29 | 0 | 0 | 0 | 0 | 30 | 96.67% |
| | Bare soil | 1 | 0 | 0 | 2 | 23 | 4 | 0 | 0 | 30 | 76.67% |
| | Other vegetation | 1 | 0 | 0 | 0 | 0 | 29 | 0 | 0 | 30 | 96.67% |
| | Standing water | 0 | 0 | 0 | 0 | 0 | 2 | 28 | 0 | 30 | 93.33% |
| | Rivers | 0 | 0 | 0 | 0 | 0 | 2 | 0 | 28 | 30 | 93.33% |
| | Totals | 34 | 22 | 26 | 32 | 24 | 46 | 28 | 28 | **240** | |
| | Producer accuracy | 85.29% | 100.00% | 100.00% | 90.63% | 95.83% | 63.04% | 100.00% | 100.00% | | |
| | Overall accuracy | 89.17% | | | | | | | | | |

## 5. Conclusions and Recommendations for Future Work

A comprehensive analysis of available satellite image archives was carried out for the 1991 to 2019 time period (1991, 1995, 2000, 2002, 2005, 2010, 2015 and 2019) covering mainly changes in terms of Armenian Forest policies. The Forest policy has had contradictory changes since Armenia's independence from the former USSR in 1991. The extensive conservative forest policy existing in Soviet Armenia since the 1950s has resulted in a large share of aged forest stands with unsatisfactory self-regeneration [20]. The first National Forest Code from 1994 was simply trying to keep Soviet era traditions in place and did not reflect the new realities. A lot of illegal and unregulated industrial logging and export of lumber in the 1990s and early 2000s were not sufficiently managed or regulated by law. In turn, the acting forest code, which was adopted in 2005 and included the newly introduced "production forest" category, was a belated response to the real situation in the forest sector. Due to these mismatched forest policies, Armenia's scarce mountain forest resources were and are undergoing severe and continuous degradation processes [21,22].

Landsat and Sentinel 2 data were processed with the support of VHSR imagery in 2005 and 2019 for calibration and validation. Several products were generated to identify:

- Forest densities for each reference years
- Forest types for the most recent year (2019)
- Land cover types for each reference year
- In addition, consistent change maps were generated to identify:
- Forest cover changes including the identification of degraded versus deforested areas
- Land cover types of changes

These provide detailed information on the nature of forest cover changes in the Dilijan National Park and what the main causes of deforestation are.

From the analysis of the results, it is shown that the forested area is primarily covered by broadleaved species and has remained relatively high over time representing about 50% of the area. Most of the deforestation and degradation occurred prior to 2000 with a relatively stable period until 2005 when both deforestation and degradation started to increase again, although this seems to have again stabilized in recent years. In addition, there also seems to have been some substantial forest regeneration from previously deforested/degraded areas.

From the land cover type maps produced, it would appear that the main causes of forest cover changes are linked to (i) conversion to other type of vegetation, (ii) agriculture and (iii) settlements, but as mentioned these changes are relatively minor.

There are no previous studies over specially protected areas in Armenia using remote sensing, hence the importance of this publication. This study should provide useful information to the management of Dilijan Park and the Armenian authorities for the effectiveness and impact of their forest policies and can serve as a model for a future nationwide forest monitoring system.

**Author Contributions:** M.S. (Michaela Seewald) was the project coordinator. A.M., C.S., M.S. (Michaela Seewald) were leading the study tasks. N.M., A.M., M.S. (Martin Siklar), N.K., L.F. participated to the realization of the study. H.S. was the client of the study and follow the project from the beginning. All authors participated to the redaction, proofreading and reviewer answers. All authors have read and agreed to the published version of the manuscript.

**Funding:** This research was funded by the European Space Agency (ESA) within the framework of the EOCLINIC project EOC0004 Characterization of Dilijan National Park Forest Ecosystems, in Armenia.

**Institutional Review Board Statement:** Not applicable.

**Informed Consent Statement:** Not applicable.

**Data Availability Statement:** Not applicable.

**Acknowledgments:** We acknowledge support given by the United Nations Development Programme (UNDP) Armenia.

**Conflicts of Interest:** The authors declare no conflict of interest.

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
