# Peer review of "Development and Application of Earth Observation Based Machine Learning Methods for Characterizing Forest and Land Cover Change in Dilijan National Park of Armenia between 1991 and 2019"

_remotesensing, doi:10.3390/rs13152942_

Round 1
Reviewer 1 Report
- This paper focused on the land cover change in the Dilijan National Park of Armenia. I think the authors could discuss the relevant studies about the monitoring of specially protected areas from remote sensing to draw wider attention, especially in the discussion section. The scientific meaning of this study and its contribution to relevant research should be highlighted.
- The figures are of low quality. For instance, Figures 4 and 6 are low in resolution. In Figure 3, some images are missing in the bottom part of the area. Figure 14: The annotations are too small. Figures 16 and 18: the two subfigures should be placed separately (not too close).
- Figure 7: why you use only red and near-infrared bands in the change detection. Since you are analyzing multiple classes, I think only two bands are insufficient.
- The Landsat, Sentinel-2, and Ikonos images were used. The authors can list the usage of each type of data at the beginning of the data description.
- Figures 19-21: use tables instead of figures to demonstrate the error matrix.
- Add more references.
Reviewer 2 Report
This paper describes a research project finaced by public fund. It produced different maps of an Armenian park. It is well written, but the description of the method for each map is relatively weak, and not all products have been validated. The introduction provides a good context, but nothing about existing methods used to achieve the same results. I am not used to review this category of paper, so I think that only minor changes are needed to fit with the requirements. The validation should however be adjusted (including the validation of the forest type map), and it should be specified more clearly when GEOBIA or pixel-based approaches are used (which also has an impact on the validation)
Considering the objective of the paper, a comparison with the global forest watch products would have been interesting (this study goes further back in time, but GFW has a yearly update).
Fig 3 (and 13) there is a white line accross the figure; As a general comment, coordinate grid is missing.
L205: Not clear if you use the method of Merciol et al as such or if it has been modified. There is indeed a different context and a different spatia resolution. The novelties should be highlighted and the reuse of methodproperly cited.
L217 : typically, more information about the method (or citation) needed here
L241 : do you mean from 10 m to 30 m ?
L274 : please explain the rule of labelling. Is it based on the central point, on the majority within the pixel, or else ?
Figure 5: What is the "other" class if you have forest/non forest. Is it "No data" ? "non vegetation" ?
L295 : If I understand well, the PSU has a resolution of 100 m but the density maps have a resolution of 30 m. How is the change of resolution managed in the project ?
L315 : how is the change "proven by EO data"?
Fig 6: the discrimination between coniferous and broadleaf seems to primarily rely on the discrimination of deciduous/sempervirens trees based on phenology. Please justify this choice based on the trees species present in the area (e.g. no Larix sp, no Buxus sp, no Sempervirens Quercus...)
L340 : it is unclear how a density map crossed with a forest type map can produce a dominant leaf type map. The leaf type dominance should be derived by the proportion of each leaf type, while the density provides information about closed canopy/open forest. What is the spatial unit of aggregation?
L384 : Why did you switch from MAJA to Sen2Cor (or the opposite)?
L399 : reference not found
L432 : citation to change vector analysis method is needed
Fig 9 : why is the change threshold different for positive and negative changes ? Please merge the two "no change" classes in the legend.
L538 : figure 14
Fig 18 : the circles are not centered on the same location and do not highlight the changes described in the caption.
L603: stratified sampling (based on classification results) was used in this study, but the sampling probability was not taken into account for the computation of producers and overall accuracies. This must be corrected.
Round 2
Reviewer 1 Report
I'm satisfied with the responses. But I still think the figures can be represented more clearly.
Author Response
We modified accordingly to the recommendation i.e. improvement of figure quality and lisibility